# Dendrogenin A Enhances Anti-Leukemic Effect of Anthracycline in Acute Myeloid Leukemia

**DOI:** 10.3390/cancers12102933

**Published:** 2020-10-12

**Authors:** Pierre-Luc Mouchel, Nizar Serhan, Rémy Betous, Thomas Farge, Estelle Saland, Philippe De Medina, Jean-Sébastien Hoffmann, Jean-Emmanuel Sarry, Marc Poirot, Sandrine Silvente-Poirot, Christian Récher

**Affiliations:** 1Service d’Hématologie, Centre Hospitalier Universitaire de Toulouse, Institut Universitaire du Cancer de Toulouse Oncopole, 31059 Toulouse, France; pierre-luc.mouchel@inserm.fr; 2Centre de Recherches en Cancérologie de Toulouse, UMR1037, Inserm, Université de Toulouse 3 Paul Sabatier, Equipe Labellisée LIGUE 2018, F-31037 Toulouse, France; nizar.serhan@inserm.fr (N.S.); thomas.farge@inserm.fr (T.F.); estelle.saland@inserm.fr (E.S.); jean-emmanuel.sarry@inserm.fr (J.-E.S.); 3Team “Cholesterol Metabolism and Therapeutic Innovations”, Cancer Research Center of Toulouse (CRCT), UMR 1037, Inserm-Université de Toulouse 3, Equipe labellisée par la ligue contre le cancer, 31037 Toulouse, France; sandrine.poirot@inserm.fr; 4CRCT, Université de Toulouse, Inserm, CNRS, UPS, 31000 Toulouse, France; remy.betous@inserm.fr; 5Equipe Labellisée Ligue Contre le Cancer, Laboratoire d’Excellence Toulouse Cancer, 31037 Toulouse, France; 6Affichem SAS, 31400 Toulouse, France; philippe.de-medina@inserm.fr; 7Laboratoire d’Excellence Toulouse Cancer (TOUCAN), Laboratoire de pathologie, Institut Universitaire du Cancer-Toulouse, Oncopole, 31037 Toulouse, France; jean-sebastien.hoffmann@inserm.fr

**Keywords:** AML, LXR, dendrogenin A, DDA, anthracycline, synergy, primary sample, CLDX, DNA damage, autophagy

## Abstract

**Simple Summary:**

Recently, several molecules have improved the clinical outcome of acute myeloid leukemia (AML) patients. Despite these recent advances, their prognosis remains poor and new strategies to improve the standard anthracycline and Ara-C-based chemotherapy are needed. We recently published that dendrogenin A (DDA), a mammalian cholesterol metabolite with tumor-suppressor properties, can potentiate the effect of Ara-C to kill AML cells. In this study, we find that DDA can also potentiate anthracycline against AML. The potentiation of Ara-C by DDA is due to a switch from a protective autophagy to a deadly autophagy. Regarding anthracyclines, the potentiation of daunorubicin is caused by the modulation of the efflux by the PgP pump, and that of idarubicin, to an increase in DNA damage and to the induction of a rapid and lethal autophagy. This is caused by rapid modulation of AKT/mTOR and JNK activity, two major pathways involved both in DNA repair and lethal autophagy.

**Abstract:**

Dendrogenin A (DDA), a mammalian cholesterol metabolite with tumor suppressor properties, has recently been shown to exhibit strong anti-leukemic activity in acute myeloid leukemia (AML) cells by triggering lethal autophagy. Here, we demonstrated that DDA synergistically enhanced the toxicity of anthracyclines in AML cells but not in normal hematopoietic cells. Combination index of DDA treatment with either daunorubicin or idarubicin indicated a strong synergism in KG1a, KG1 and MV4-11 cell lines. This was confirmed in vivo using immunodeficient mice engrafted with MOLM-14 cells as well as in a panel of 20 genetically diverse AML patient samples. This effect was dependent on Liver X Receptor β, a major target of DDA. Furthermore, DDA plus idarubicin strongly increased p53BP1 expression and the number of DNA strand breaks in alkaline comet assays as compared to idarubicin alone, whereas DDA alone was non-genotoxic. Mechanistically, DDA induced JNK phosphorylation and the inhibition of AKT phosphorylation, thereby maximizing DNA damage induced by idarubicin and decreasing DNA repair. This activated autophagic cell death machinery in AML cells. Overall, this study shows that the combination of DDA and idarubicin is highly promising and supports clinical trials of dendrogenin A in AML patients.

## 1. Introduction

The recently discovered Dendrogenin A (DDA) is the first tumor-suppressor of cholesterol origin [1,2]. DDA is present in normal mammalian tissues and displays significant preclinical anticancer activity in several cancers including acute myeloid leukemia (AML) [3,4]. DDA metabolism is deregulated during oncogenesis, since its levels are dramatically decreased in tumors, and it was not detected in a panel of solid tumor and AML cell lines [1]. DDA is an inhibitor of cholesterol epoxide hydrolase (ChEH), the enzyme that catalyzes the hydrolysis of 5,6α- and 5,6β-epoxy-cholesterol (5,6-ECs) into cholestan-3β,5α,6β-triol (CT) [5]. ChEH activity is carried out by two enzymatic subunits involved in cholesterol synthesis, namely the 3β-hydroxysterol-Δ8,7-isomerase (D8D7I or EBP) and the 3β-hydroxysterol-Δ7-reductase (DHCR7) [6]. We recently showed that DDA selectively triggers the lethal autophagy of AML cells in vitro and in vivo through a distinctive mechanism that involves the direct activation of the nuclear receptor transcription factor liver-X-receptor beta (LXRβ) and the inhibition of the cholesterogenic function of D8D7I [3]. The anti-leukemic activity of DDA was independent of cytogenetic or molecular AML subtypes and prominent in CD34^+^CD38^−^CD123^+^ AML cells, indicating a broad potential for this compound in the therapy of AML patients. In this study, we evaluated the activity of the DDA-anthracycline combination in AML to provide scientific rationale for future clinical trials.

## 2. Results

### 2.1. Daunorubicin Combined with DDA Synergistically Induce AML Cell Death 

We first evaluated the cytotoxic activity of the DDA plus daunorubicin combination in AML cell lines in vitro. Data from the dose-effect analysis were used to calculate the combination index (CI) using the equation of Chou and Talalay [7]. This assessment of multiple drug effect interactions indicated a strong synergism between the two drugs in KG1a, KG1 and MV4-11 cell lines (Figure 1A). We next investigated whether this synergistic effect could be due to the modulation of daunorubicin efflux upon DDA treatment. For this, we used the KG1a cell line, which displays a very potent drug efflux through P-glycoprotein (Pgp) activity [8]. Following a 24-hour pretreatment with DDA or vehicle, KG1a cells were incubated with daunorubicin for 30 min, and its intrinsic fluorescence was assessed by flow cytometry after an additional 30 min to allow for drug release. Under these conditions, DDA strongly increased the intracellular retention of daunorubicin in KG1a cells compared to control (Figure 1B and Appendix A). In line with this result, Pgp surface expression was downregulated upon DDA treatment in four primary samples from AML patients (Figure 1C). Both daunorubicin efflux and inhibition of Pgp expression were no longer observed in KG1 cells in which the major target of DDA, namely LXRβ, was downregulated by a specific Sh-RNA (Figure 1D,E and Appendix A). This suggests that DDA-induced downregulation of PgP expression and activity is LXRβ-dependent. Idarubicin was observed to be less sensitive to Pgp activity than daunorubicin [9].

### 2.2. Idarubicin Combined with DDA Synergistically Induce AML Cell Death 

Using the same conditions as described above with daunorubicin, the pretreatment of KG1a cells with DDA did not affect the intracellular concentration of idarubicin (Figure 1F). However, the DDA plus idarubicin combination was still synergistic in KG1a cells (Figure 2A). DDA also increased the toxicity of idarubicin in MOLM14 and OCI-AML3 cell lines (Figure 2B,C) and this effect was also LXRβ-dependent (Figure 2D and Appendix A). The activity of this drug combination was also addressed in vivo using an immunodeficient mice model engrafted with MOLM-14 cells subcutaneously or orthotopically via tail vein injection. Compared to single treatments with DDA or idarubicin alone, the combination significantly reduced the total tumor cell burden (Figure 2E,F). The cytotoxicity of the DDA plus idarubicin combination was also assessed in primary samples from 20 AML patients (Appendix A; Figure 2G). Cell survival was reduced by 42% in DDA–idarubicin-treated cells compared to cells treated only with DDA, 50% when compared to idarubicin-treated cells and 68% when compared to vehicle. DDA–idarubicin cytotoxicity was observed regardless of the cytogenetic, molecular or chemosensitivity status of the AML samples (Table 1 and Figure 2H–J). Altogether, these data indicate that DDA increased anthracycline activity at least in part through a PgP-independent mechanism. Moreover, DDA did not increase idarubicin cytotoxicity in normal PBMC and lymphocytes in vitro and had no effect on normal murine hematopoiesis in vivo, indicating a potential favorable benefit/risk ratio (Appendix A).

### 2.3. Dendrogenin A Increases DNA Damage Induced by Idarubicin Treatment

Because DDA increases idarubicin cytotoxicity independently of drug efflux, unlike daunorubicin, we investigated the impact of the DDA plus idarubicin combination on the DNA damage response. p53BP1 expression, a specific and early marker of DNA double-strand breaks [10,11], was assessed by immunofluorescence in four primary AML samples treated with DDA, idarubicin or DDA plus idarubicin. Whereas DDA did not induce p53BP1 expression, the DDA plus idarubicin combination strongly increased p53BP1 expression as compared to idarubicin alone (Figure 3A,B). To confirm this result, alkaline comet assays were performed using a new highly reproducible technology to detect DNA single- and double-strand breaks: the 4D Lifetest^TM^ 1.0 developed by 4D lifetec, Cham, Switzerland. DNA breaks induced by idarubicin were significantly increased by DDA compared to idarubicin alone in both primary AML samples (*n* = 5) and three different cell lines, whereas DDA alone showed no genotoxic effect (Figure 3C–F). Altogether, these results indicate that the DDA–idarubicin combination strongly enhances the DNA targeting of anthracyclines.

### 2.4. DDA–Idarubicin-Induced Cell Death is Mediated by Autophagy

To investigate whether other mechanisms could be involved in the DDA–idarubicin synergy, we sought to determine which type of cell death mechanism is activated following the combination treatment. We first treated KG1a and OCI-AML3 cell lines with a caspase inhibitor (Z-VAD-FMK) before DDA–idarubicin treatment. Z-VAD-FMK had no effect on cell death induced by this combination, suggesting that the synergism between DDA and idarubicin is not dependent on an apoptotic mechanism (Figure 4A).

We have previously shown that DDA induces cell death through a lethal autophagic process [3,4]. We therefore investigated the role of autophagy in the cell death induced by DDA–idarubicin combination. Primary AML cells treated with DDA–idarubicin showed a massive accumulation of cytoplasmic macrovesicles visualized by MGG staining (Figure 4B) and increased in MOLM-14, OCI-AML3 and KG1a cell lines [12,13], compared to each drug alone, the levels of LC3-II, which is the only protein marker that is reliably associated with completed autophagosomes [14]. Moreover, the pharmacological inhibition of autophagy using the lysosomal protease inhibitor bafilomycin A1 (BafA1) abolished the synergism between DDA and idarubicin in these cell lines (Figure 4D–F; Appendix A). To further explore the role of autophagy in this model, we transduced KG1 cells with a lentiviral vector encoding inducible shRNA against VPS34, a key autophagy protein [14]. Following VPS34 knock down, the synergism between DDA and idarubicin was also abrogated (Figure 4G). These results strongly suggest that DDA potentiates Idarubicin-induced cell death through a rapid and lethal autophagic mechanism.

It has been recently shown that a strong inhibition of the AKT/mTOR pathway concomitant to c-Jun N-terminal kinase (JNK) activation occurs when cells are unable to repair their DNA damage, and leads to lethal autophagy [15]. Interestingly, after a short-term (5 h) incubation with DDA, idarubicin or DDA–idarubicin, the combination treatment strongly downregulated the phosphorylation of AKT induced by idarubicin and increased the phosphorylation of JNK in OCI-AML3 cells but did so less markedly in the KG1a cell line. This therefore suggests a link between the DNA damage response induced by the DDA–idarubicin combination and the mechanism of cell death by lethal autophagy (Figure 4H,I and Appendix A).

Altogether, our preclinical work provides a scientific rationale for future clinical trials of DDA-anthracycline combinations in AML.

## 3. Discussion

In this study, we showed that the DDA–anthracycline combination exhibited a highly synergistic cytotoxic activity in AML cells compared to normal hematopoietic cells. This effect was associated with a strong induction of DNA damage and cell death by autophagy. Furthermore, the DDA–idarubicin combination rapidly modulated the activation of AKT/mTOR and JNK, two major pathways involved in both DNA repair and lethal autophagy.

Anthracyclines are a major component of the backbone treatment in AML and the intensification of daunorubicin doses has been pivotal in establishing the standard induction treatment [16]. The two main anthracyclines used in AML (i.e., daunorubicin and idarubicin) intercalate into DNA and form complexes with topoisomerase II, thereby inhibiting the activity of the enzyme and inducing single- or double-stranded breaks that lead to a subsequent DNA-damage response and cell death [17,18]. Anthracyclines also elicit multiple signaling events, including activation of the ROS-dependent sphingomyelin-ceramide pathway [19], a very early event following anthracycline treatment and a crucial step in JNK pathway activation and cell death [20]. Several mechanisms of resistance to anthracyclines, including the so-called multi-drug resistance (MRD) phenotype, have been described [21]. In our study, we demonstrated that DDA, which does not induce DNA damage by itself, dramatically enhanced the number of DNA strand-breaks induced by idarubicin. The mechanisms underlining this observation remain unclear. However, recent data have shown that autophagy may impact genome integrity through the degradation of nuclear components and modulation of the double-strand break repair pathway [22,23]. For example, the activation of autophagy has been shown to induce the degradation of the DNA endonuclease Sae2, thereby decreasing cell survival in response to the genotoxic agent [24]. Here, DDA, which is a very potent autophagy-inducing agent, could impact the DNA damage response following anthracycline exposure by inducing the degradation of nuclear components involved in anthracycline activity.

We previously showed that the combination of DDA, which induces cell death through lethal autophagy, with the antimetabolite ARAc, which induces mainly apoptosis, synergized in killing AML cells [3,4]. Here, we show that the combination of DDA with a DNA-damaging agent potentiates AML cell-killing, showing that the multiple-cell-death mechanism improves the yield of AML cell-killing, supporting the AL-lazikani’s hypothesis [25]. Since DDA has been proved to be efficient on various cell lines in vitro and in vivo [26,27], this suggests that combination treatment of DDA with chemotherapeutic agents that kill cells through other mechanisms could be as efficient and deserves further investigation.

Further studies are needed to determine which molecules of the DNA repair machinery might be targeted by DDA treatment, either through autophagy or its LXR-β dependent transcriptional activity [28]. We showed that DDA induced JNK phosphorylation and the inhibition of AKT phosphorylation in AML cell lines. It has recently been shown that DNA damage may regulate autophagy through these signaling pathways [15]. The AKT/mTOR axis may be activated following treatment with DNA-damaging agents and suppress autophagy while regulating DNA repair and cell survival [29]. On the other hand, the sustained activation of JNK following DNA damage could activate the first step of autophagy through the transcription of autophagy-related genes and through direct activation of the Beclin 1 complexes [30] and contribute to autophagic cell death.

## 4. Materials and Methods

### 4.1. Combinaison Index DDA-Daunorubicin

Experiments were designed to fit with methods previously described by Chou and Talalay to quantify the synergy of drug combinations. Briefly, the effect of drugs alone and in combination were determined for a 3–5 log scale of concentrations of drugs and with a constant ratio between the two drugs used for co-treatment. Cells were seeded in 96-well plates (50,000 cell/well). Cells were treated just after seeding, for 48 h. Cell viability was determined with MTT assay. Cells were centrifuged for 5 min at 1200 rpm, medium removed, 100 µL of MTT solution (1 mg/mL in PBS) added, incubated for 2 h at 37 °C, centrifuged, MTT solution removed, and DMSO (100 µL) added to dissolve purple formazan formed by living cells. Absorbance of the solution was determined at 540 nm.

### 4.2. Antibodies and Reagents

Anti-LC3B, anti-VPS34, anti-JNK, pJNK, AKT, pAKT antibodies were obtained from Cell Signaling Technology, Massachusetts, USA; secondary antibodies labeled with horseradish peroxidase were purchased from Promega, Madison, WI. The apoptosis inhibitor ZVAD-FMK was purchased from Merk, Darmstadt, Germany. BafilomycinA1 was purchased from InvivoGen, San Diego, CA, USA.

### 4.3. Cell Lines and AML Primary Samples

Human AML cell lines MOLM14, KG1, KG1a and OCI-AML3 were purchased from the Leibniz Institute DSMZ-German Collection of Microorganisms and Cell cultures (Braunschweig, Germany). OCI-AML3 and MOLM14 Cell lines were grown in RPMI-1640 medium with Glutamax supplemented with 10% fetal bovine serum (Dominique Dutscher, Brumath, France) while KG1 and KG1a cells were grown in Iscoves-modified Dulbecco medium (IMDM) containing 20% fetal bovine serum. Frozen samples from AML patients have been obtained after informed consent and stored at the HIMIP collection (BB18 0033-00060). According to the French law, HIMIP collections has been declared to the Ministry of Higher Education and Research (DC 2008-307 collection 1) and obtained a transfer agreement (AC 2008-129) after approbation by the “Comité de Protection des Personnes Sud-Ouest et Outremer II” (ethical committee). Peripheral blood or bone marrow samples were frozen in fetal calf serum with 10% DMSO and stored in liquid nitrogen. AML primary samples were cultured in IMDM containing 20% fetal bovine serum and 100 units/mL of penicillin and streptomycin. For some experiments, fresh leukemic blasts recovered at diagnosis were immediately treated with ethanol, idarubicin 10 nM, DDA 2.5 µM or both DDA and idarubicin for 48 h. Patient’s characteristics are shown in Appendix A.

### 4.4. Trypan Bleu Viability Test

Cell viability was determined using trypan bleu exclusion test. Briefly, 250,000 cells were either treated with ethanol, idarubicin, DDA or both DDA and idarubicin. Cells were homogenized and 10 µL of resuspended cells was mixed with 10 µL of trypan bleu solution. AML cells were visually examined with a microscope to determine whether cells take up (dead cells) or exclude dye (viable cells).

### 4.5. Chemotherapy Efflux Test 

AML cells were pre-incubated with DDA at 2.5 µM during 24 h. Then, cells were incubated with daunorubicin (1 mM) or idarubicin (200 µM), for 30 min at 37 °C. After incubation, cells were washed with cold PBS and re-suspended in IMDM medium containing 20% FBS for 30 min. Next, cells were washed with cold PBS and re-suspended in 200 µL of PBS. Intracellular concentration of daunorubicin and idarubicin was determined using flow cytometry by quantifying the fluorescent in PE-channel using a FACSCalibur flow cytometer (BD Pharmingen, San Diego, CA, USA).

### 4.6. PgP Membrane Expression

AML cell lines and primary AML samples were treated during 24 or 48 h with 2.5 µM DDA. After a cold PBS wash, cells were re-suspended in 200 µL of PBS and membrane expression of PgP was determined by flow-cytometry using CD243 from BD biosciences (Franklin Lakes, NJ, USA).

### 4.7. Apoptosis Quantitation by Flow Cytometry

Apoptosis was analyzed by flow cytometry following double staining with annexinV fluorescein isothiocyanate (FITC) and 7AAD. Briefly, 250,000 or 500,000 cells for primary cells were either treated with ethanol, idarubicin, DDA or DDA and idarubicin. Cells were collected and washed once with PBS and once with annexinV-binding buffer. After centrifugation, the pellets were incubated 15 min with annexinV FITC and 7AAD in 100 µL annexin-binding buffer. Cells were then analyzed by flow cytometry.

### 4.8. Western Blot 

Proteins were separated using 4–12% gradient polyacrylamide SDS gels (Life Technologies, Darmstadt, Germany) and electro-transferred to a 0.45 µM PVDF membrane (GE Healthcare, Chicago, IL, USA). After blocking in Tris-buffered saline containing 0.1% Tween (TBST) and 5% bovine serum albumin, membranes were blotted overnight at 4 °C with the appropriate primary antibodies. Primary antibodies were detected using appropriate horseradish peroxidase-conjugated secondary antibodies. Immunoreactive bands were visualized by enhanced chemiluminescence (Clarity ECL BIORAD, Hercules, CA, USA) with a Syngene camera. Densitometric analyses of immunoblots were performed using the GeneTools software.

### 4.9. Lentiviral Infection of KG-1

Lentiviral particles were generated by calcium phosphate transient transfection in 293T cells, as already described [4]. Briefly, 293T into a 10 cm dish were transfected with 62.5 µL CaCl2 (2M), 500 µL HeBS 2×, 418 µL H_2_O, 3.5 µg pVSV-G (env), 6.5 µg p8.1 (tat, pol, rev, gag) and 10 µg inducible shRNA against VPS34 (TRIPZ Human PIK3C3, clone V3THS_372038, GE Healthcare, Chicago, IL, USA). Then, 72 h after cell transfection, 2 mL of supernatants containing virus were collected and were added to KG-1 in a 6-well plate. Polybrene was added at 8 µg/mL final concentration and spinoculation was performed by centrifuging cells 45 min at 800 g. Seventy-two hours after transduction, medium containing virus was removed and changed for a virus-free medium. After additional 24 h, cells were selected with 1 µg/mL puromycin. When puromycin-resistant cells appeared, KG-1 expressing a high level of shRNA (RFP positive cells) were sorted by flow cytometry after 24 h treatment with 1 µg/mL doxycycline. All shRNA experiments were performed on the cell bulk, treated or not 72 h with 1 µg/mL doxycycline for shRNA induction.

### 4.10. May-Grünwald Giemsa Staining (MGG)

AML cells were treated with ethanol or idarubicin +/− DDA for 48 h. For morphological analysis, 100,000 cells in 500 µL were cytospined on slides by centrifugation at 800 RPM for 10 min. Slides were stained at room temperature with MGG and cellular morphology was examined using light microscopy.

### 4.11. Immunofluorescence

AML cells (10^5^) were treated with ethanol or idarubicin +/− DDA during 15 h and cytospined as described for MGG. Cells were then fixed with 4% paraformaldehyde during 15 min at RT. After fixation, cells were washed in PBS and Click-IT reaction was performed using Alexa647 kit from Life Technologies. Subsequently, cells were blocked with 5% BSA in PBS for 20 min. Cells were incubated overnight at 4 °C with primary antibodies against P-53BP1 (1/200, Cell signalling technologies Beverly, MA, USA) in PBS. Then, cells were washed with PBS, and then incubated with Alexa Fluor 488 goat anti–mouse (1/1000; 95 Molecular Probes) for 1 h at RT in PBS. DNA was counterstained with DAPI and coverslips mounted on microscopy-slides using ProLong Diamond (Thermofisher, Waltham, MA, USA). Images were taken with a Nikon Ni-E microscope and a DS-Qi2 camera at the 20× objective. Fluorescence intensity was measured with the Cellprofiler software (www.cellprofiler.org).

### 4.12. Alkalin Comet Assay

AML cells were mixed with low melting agarose (0.5% final) and then spread on microscopic slides from 4D lifetec, Cham, Switzerland. Slides were brought to 4 °C for about 15 min to allow solidification of the gel. Samples were lysed for 1 h at 4 °C in lysis buffer (2.5 M NaCl, 10 mM Tris, 10%DMSO and 1% triton X-100) and then rinsed in electrophoresis buffer (300 mM NaOH, 1 mM EDTA) for 30 min at 8 °C. Electrophoresis was then run using the 4D Lifetest^TM^ 1.0 (4DLifetec) at 20 V for 40 min at 8 °C. Slides were incubated in PBS for 15 min and 1 h 30 in 100% ethanol at 4 °C. After drying at 37 °C for 15 min, DNA was stained with SYBR^®^Gold during 15 min. The slides were then rinced twice with deionized water during 10 min. Finally, slides were dried again prior to imaging with a Nikon Ni-E microscope and a DS-Qi2 camera at the 10× objective. Comet tail intensity was quantified using the casplab software 1.2.2 (http://casplab.com).

### 4.13. Xenograft Model and In Vivo Antitumor Activity Assay

As previously described [4,31,32] NOD/LtSz-scid (NOD/SCID) and NOD/LtSz-scid/IL-2Rγchain null (NSG) mice were produced at the Genotoul Anexplo platform (Toulouse, France) using breeders obtained from Charles River Laboratory. Animals were used in accordance with a protocol reviewed and approved by the Institutional Animal Care and Use Committee of Région Midi-Pyrénées 9001v2008 ISO certified in June 2010 (Toulouse, France). For tumor xenograft experiments, Molm14 (2 × 10^6^) were inoculated subcutaneously into the flanks of NOD/SCID mice. When tumors were palpable, mice were treated. Tumor volume (tumor length × width² × 0.5236) was measured every 2–3 days using calipers. At the end of the experiment, or when animals show bad health condition, mice were humanely sacrificed. For orthotopic xenograft, AML cell lines were transplanted into NSG mice, as reported previously [4,7,8]. Briefly, mice were housed in sterile conditions using HEPA-filtered micro-isolators and fed with irradiated food and sterile water. Transplanted mice were treated with antibiotic (baytril) for the duration of the experiment. Mice (6–9 weeks old) were sublethally treated with busulfan (20 mg/kg) 24 h before injection of leukemic cells. Leukemia samples were thawed at room temperature, washed twice in PBS, and suspended in phosphate saline buffer at a final concentration of 1 million cells per 100 μL. A total of 200 µL of phosphate saline buffer solution containing 2 × 10^6^ of AML cells were injected in tail vein. Five days after AML cells transplantation, NSG recipients transplanted mice were injected in blood tail with 0.15 mg/kg of idarubicin every two days for 5 days and/or 20 mg/kg/day of DDA 5 days/7 during two weeks. For negative controls, NSG mice were treated daily with IP injection of vehicle, PBS 1×. Mice were monitored for toxicity and provided nutritional supplements as needed. NSG mice were humanely killed in accordance with European ethic protocols. Bone marrow (mixed from tibias and femurs) and spleen were dissected in a sterile environment and crushed in Hanks balanced salt solution with 1% FBS. Mononuclear cells from bone marrow and spleen were labeled with FITC-conjugated anti-hCD3, PE-conjugated anti-hCD33, PerCP-Cy5.5-conjugated anti-mCD45.1, APC-conjugated anti-hCD45 and PeCy7-conjugated anti-hCD44 (all antibodies from BD, except FITC-conjugated anti-hCD3 from Biolegend, San Diego, CA, to determine the fraction of human AML blasts (hCD45+hCD33+mCD45.1- cells) using flow cytometry. Analyses were performed on a Beckman coulter cytoflex flow cytometer. The number of AML cells/µL peripheral blood and number of AML cells in total cell tumor burden (in bone marrow and spleen) were determined by using CountBright beads (Invitrogen, Waltham, MA, USA) using the described protocol.

### 4.14. Statistical Analyses

We assessed the statistical analysis of the difference between 2 sets of data using non-parametric Mann–Whitney test or Welch *t*-Test or ANNOVA for experiment including more than two sets of data (GraphPad Prism, GraphPad Software, La Jolla, CA). *p* values of less than 0.05 were considered to be significant (* *p* < 0.05, ** *p* < 0.01 and *** *p* < 0.001, **** *p* < 0.0001).

## 5. Conclusions

In this study, we report that the newly identified tumor suppressor and cholesterol metabolite DDA potentiated and sensitized in vitro and in vivo leukemic cells lines and primary AML samples to anthracycline, which is one of the two drugs used in the backbone of AML treatment. Here, we demonstrate that DDA strongly increased the intracellular retention of daunorubicin but not idarubicin by downregulating the PgP expression and activity through LXRβ. DDA, which has no genotoxic effect, significantly increased the DNA damage induced by idarubicin in cell lines and primary AML sample, indicating that the DDA–idarubicin combination strongly enhances DNA targeting of anthracycline and is not mediated by a variation in the intracellular retention of Idarubicin. Autophagy is a well-known mechanism of cell survival that limits the efficacy of conventional chemotherapy such as anthracycline. We demonstrate that a ligand of LXRβ, the DDA, can potentiate Idarubicin-induced cell death through a rapid and lethal autophagy by rapidly modulating the activation of AKT/mTOR and JNK, two major pathways involved in both DNA repair and lethal autophagy in vitro and in vivo. DDA did not increase idarubicin cytotoxicity in normal PBMC or in healthy lymphocytes from AML patients in vitro. Moreover, DDA had no effect on normal murine hematopoiesis in vivo, indicating a potential favorable benefit/risk ratio.

A DNA damaging agent such as idarubicin with DDA creates the necessary conditions to maximize DNA damage, reduce DNA repair and activate the autophagic cell death machinery in AML cells. This provides a strong rationale to assess further this combination in early phase clinical trials for AML patients.

## Figures and Tables

**Figure 1 cancers-12-02933-f001:**
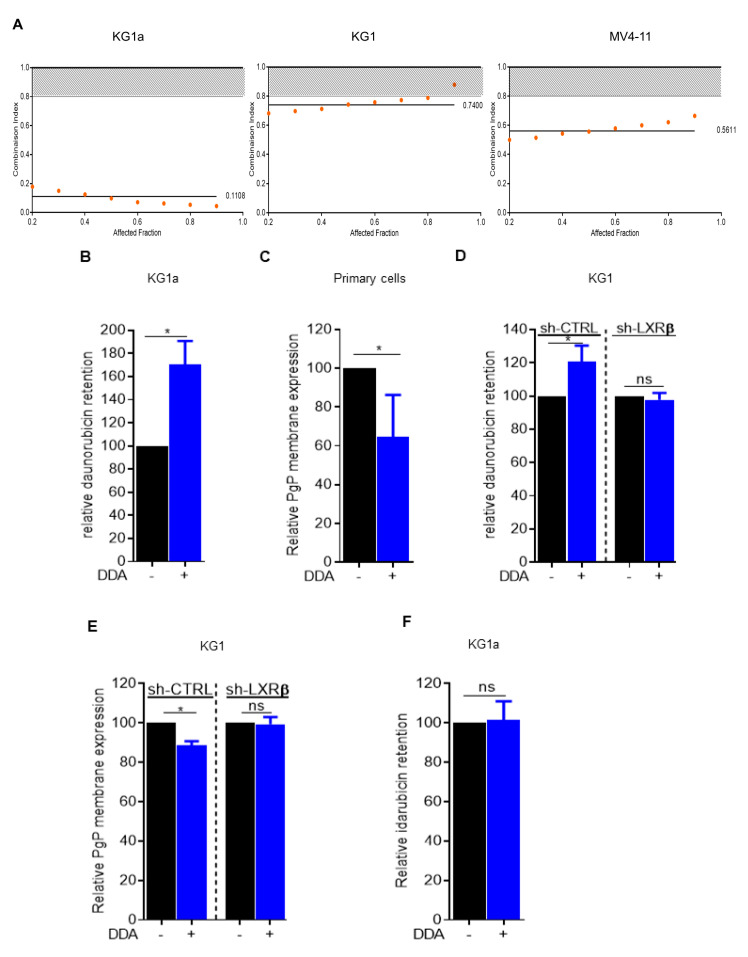
Combined dendrogenin A (DDA) and daunorubicin synergistically induce the death of AML cells. (**A**) Combination Index: CI < 0.80 indicates synergy of DDA with daunorubicin in KG1a, KG1, MV4-11 cells (**B**) Intra-cellular daunorubicin concentration was quantified by flow cytometry after 24 h of DDA treatment at 2.5 µM. DATA were represented as % of intra-cellular daunorubicin concentration relative to vehicle treated cells. Bars are mean ± SD of 3 independent experiments. (**C**) PgP membrane expression was measured by flow cytometry after 48 h of DDA treatment at 2.5 µM in four primary AML sample. Data were represented as % of PgP membrane expression relative to vehicle treated cells. (**D**) Intra-cellular daunorubicin concentration was quantified after 24 h of DDA treatment at 2.5 µM in KG1-shctrl and Kg1-ShLXRβ cells. Data were represented as % of intra-cellular daunorubicin concentration relative to vehicle treated cells. Points are mean ± SD of 3 independent experiments. (**E**) PgP membrane expression was measured after 24 h of DDA treatment at 2.5 µM in KG1Shctrl and KG1-ShLXRβ. Data were represented as % of PgP membrane expression relative to vehicle treated cells. Bars are mean ± SD of 3 independent experiments. (**F**) Intra-cellular IDA concentration was quantified by flow cytometry after 24 h of DDA treatment at 2.5 µM. Data were represented as % of intra-cellular idarubicin concentration relative to vehicle treated cells. Bars are mean ± SD of 3 independent experiments. * *p* < 0.05.

**Figure 2 cancers-12-02933-f002:**
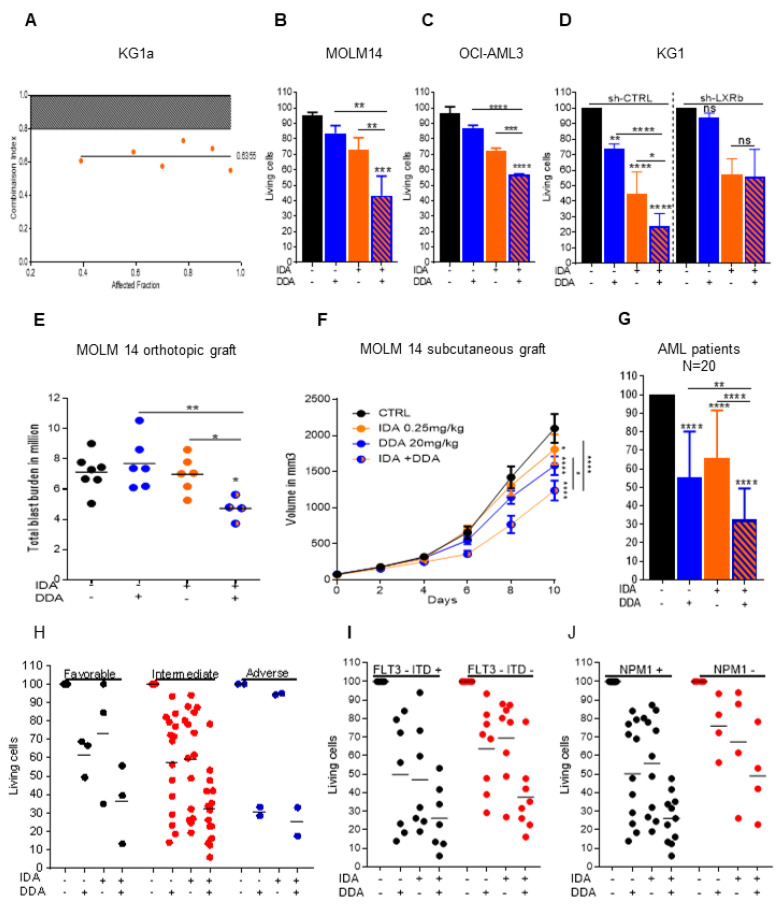
Combined DDA and idarubicin synergistically induce the death of AML cells (**A**) Combination Index: CI < 0.80 indicates synergy of DDA with idarubicin in KG1a cells. (**B**,**C**) Cell death measurement by Trypan Blue exclusion test on MOLM-14 and OCI-AML3 cells treated for 48 h with DDA (2.5 µM), IDA at 25 nM for MOLM-14 and 50 nM for OCI-AML3 or both DDA and idarubicin at the same concentration. Bars are mean ± SD of 3 independent experiments. (**D**) Cell death measurement by flow cytometry on KG1-ShCTRL and KG1-ShLXRβ treated for 48 h with DDA (5 µM), IDA (50 nM) or both DDA and IDA (5 µM/50 nM). Data were represented as % of survival corresponding to annexin-V-/7AAD- cells relative to total cells. Bars are mean ± SD of 3 independent experiments. (**E**) Total tumor burden of MOLM14 orthotopically xenografted NSG mice were treated with DDA (20 mg/kg/day by i.p injection) or IDA (0.15 mg/kg/day every two days for 5 days by i.v injection) or both DDA and IDA or vehicle control. MOLM-14 leukemic cell burden in bone marrow and spleen was measured by flow cytometry using human anti-CD45, anti-CD45.1 and human anti-CD33 antibodies. (**F**) Tumor volume curve of MOLM14 xenografts (*n*= 15 per group) in NOD/SCID mice treated with DDA (20 mg/kg/day by i.p injection) or IDA (0.25 mg/kg/day every two days for 5 days by i.v injection) or both DDA and IDA or vehicle control. G-J) Samples from AML patients (*n* = 20) were treated with DDA (2.5 µM) or IDA (10 nM) or both DDA and IDA (2.5/10 nM) or vehicle for 48 h. Cell death was assessed in the leukemic bulk (CD45+) using annexinV/7AAD staining. Data were represented as % of survival corresponding to annexin-V-/7AAD- cells relative to total cells for all AML patients (**G**) or relative to their prognostic risk category (Low (LR), intermediate (IR), and high risks (HR)) (**H**) or according to FLT3-ITD (**I**) or NPM1 status (**J**). * *p* < 0.05, ** *p* < 0.01 and *** *p* < 0.001, **** *p* < 0.0001.

**Figure 3 cancers-12-02933-f003:**
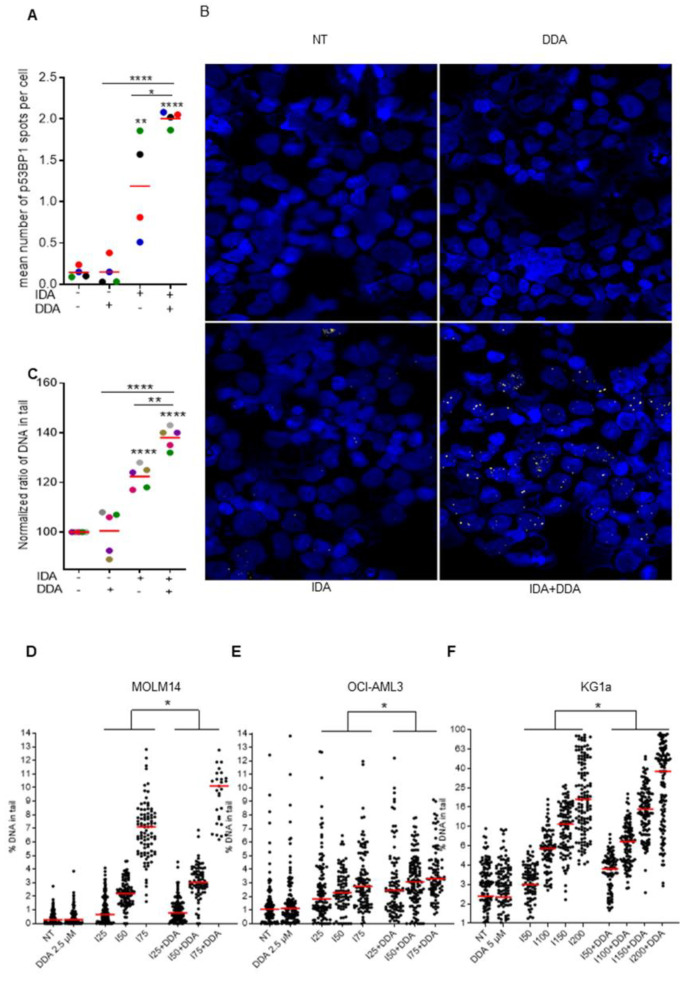
DDA potentiates DNA damage due to idarubicin (**A**) Sample of AML patients (*n* = 4) were treated with DDA (2.5 µM) or IDA (10 nM) or both DDA and IDA (2.5/10 nM) or vehicle for 15 h and then analyzed in immunofluorescence. Data were represented as mean of p53BP1 spots per nuclei. (**B**) Representative images of primary AML cells. Nuclei are represented in blue and p53BP1 appear in red. (**C**) Sample of AML patients (*n* = 5) were treated with DDA (2.5 µM) or IDA (10 nM) or both DDA and IDA (2.5/10 nM) or vehicle for 15 h and then a standardize comet assay was performed. Data were represented as mean % of DNA in tail in cells relative to the vehicle-treated cells. (**D**–**F**) AML cell lines (MOLM14, OCI-AML3 and KG1a) were treated with vehicle, DDA (2.5 µM or 5 µM) or IDA at increase concentration or both DDA and IDA for 15 h and then a standardized comet assay was performed. Data were represented as % mean of DNA in tail in cells. Spots represent the % of DNA in tail in one single cell of 3 independent experiments. * *p* < 0.05, ** *p* < 0.01 and *** *p* < 0.001, **** *p* < 0.0001.

**Figure 4 cancers-12-02933-f004:**
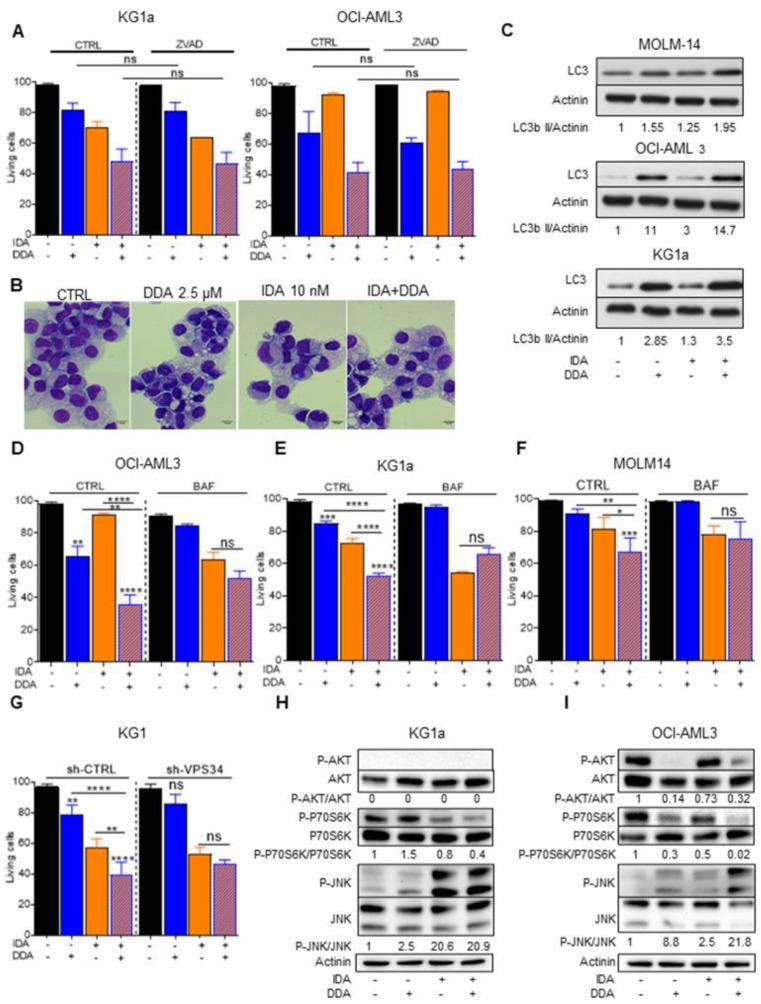
Implication of autophagy after DDA–idarubicin (IDA) treatment (**A**) Cell death measurement by Trypan Blue exclusion test after caspase inhibition on KG1a and OCI-AML3 cells treated for 48 h with DDA, IDA or both DDA and IDA in the presence or absence of ZVAD-FMK (40 µM). Bars are mean ± SD of 3 independent experiments. (**B**) Representative images of primary AML cells stained with May Grumwald Giemsa after treatment with DDA (2.5 µM) or IDA (10 nM) or both DDA and IDA (2.5/10 nM) or vehicle. (**C**) Western blot analysis of the expression of LC3-b II in three AML cell lines (MOLM-14, OCI-AML3 and KG1a) treated 5 h with vehicle, DDA, IDA or both IDA and DDA. (**D**–**F**) Cell death measurement by Trypan Blue exclusion test after autophagy inhibition on OCI-AM3, KG1a and MOLM14 cells treated for 48 h with DDA, IDA or both DDA and IDA in the presence or absence of bafilomycin A1. Bars are mean ± SD of 3 independent experiments. (**G**) Cell death measurement by Trypan Blue exclusion test after autophagy inhibition on KG1 shVPS34 treated for 48 h with DDA, IDA or both DDA and IDA. Bars are mean ± SD of 3 independent experiments. (**H**,**I**) Western blot analysis of the expression of p-AKT, AKT, pP70S6K, P70S6K, JNK and pJNK in two AML cell lines KG1a and OCI-AML3) treated for 5 h with vehicle, DDA, IDA or both IDA and DDA. Detailed information can be found at Appendix A. * *p* < 0.05, ** *p* < 0.01 and *** *p* < 0.001, **** *p* < 0.0001.

**Table 1 cancers-12-02933-t001:** Molecular and cytogenetic characteristics of primary acute myeloid leukemia (AML) cancer cells from 20 patients.

Samples	Karyotype	FLT3-ITD	FLT3-TKD	NPM1	CEBPA	IDH1	IDH2	DNMT3A
AML1	46, XY<20>	1	0	1	0	0	0	1
AML2	46, XX<20>	1	0	1	0	0	0	0
AML3	46, XX<20>	0	1	1	0	0	0	0
AML4	46, XX, del(20)(q11q13)<20>	0	0	0	0	0	1	0
AML5	46, XX<20>	0	0	1	0	0	0	1
AML6	46, XX, t(1;16)(010;q10),inv(3)(014q24)<20>	ND	ND	ND	ND	ND	ND	ND
AML7	46, XX<20>	1	0	1	0	0	0	1
AML8	46, XY, r(7)<16>	ND	ND	ND	ND	ND	ND	ND
AML9	46, XX<20>	1	0	1	0	0	0	0
AML10	46, XX<20>	1	0	0	0	1	1	0
AML11	46, XY<22>	0	1	1	0	0	0	0
AML12	46, XX<20>	1	0	1	0	0	0	1
AML13	46, XY<20>	1	0	0	0	0	0	0
AML14	46,XX,+8<1>/47,der(11)t(1;11)(q24;q24)<1>/48,sd1,+8<2>/46,XX<7>	0	0	1	0	0	0	0
AML15	46, XY<20>	0	0	1	0	0	0	0
AML16	46, XY<20>	0	0	0	1	0	0	0
AML17	46, XY, inv(16)(p13q22)<10>/46,XY<10>	ND	ND	ND	ND	ND	ND	ND
AML18	46, XX<22>	0	0	1	0	0	0	0
AML19	47, XY, inv(16)(p13q22),+22<20>	ND	ND	ND	ND	ND	ND	ND
AML20	47, XY, +8, inv(16)(p13q22)<20>	ND	ND	ND	ND	ND	ND	ND

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
