# Peer review of "Dendrogenin A Enhances Anti-Leukemic Effect of Anthracycline in Acute Myeloid Leukemia"

_cancers, 2020, doi:10.3390/cancers12102933_

Round 1

Reviewer 1 Report

Interesting study on the anti-leukemic effects of a combination therapy associating dendrogenin A (DDA) -a recently described anti-cancer agent- and an anthracycline (daunorobicin or idarubicin). The synergistic effects obtained using a novel agent with conventional drugs is a plus.  A similar manuscript describing the combination of DDA with cytarabine (ARA-C) has just been published in Cancers by the same authors. Although similar, this study brings enough novelty to be published independently. The molecular mechanisms described are of great interest in the field and beyond. There are however a few mandatory points that need to be addressed to justify the hypothesis and conclusions of the manuscript.

Major points

1- Statistics were done using Mann-Whitney or t-test. This is Ok when comparing 2 sets of data, but many experiments include more than 2 sets of data. In those cases, these cannot be used. Use ANNOVA instead, otherwise, the multiple comparisons do not stand.

2- Use classical standard deviation (SD) not SEM throughout the figures. Unlike SD, SEM is not a descriptive statistics (does not describe the variability of measures). SEM is only used here to minimize the error bar because SEM is smaller than SD).

3- Figure 2E is a quantification of human MOLM14 cells in mice. The text and legend state that the blast cells were quantified in bone marrow and spleen. The figure should shows two quantification (bone marrow and spleen), why is there only one?

4- Figure 2E: why are the groups imbalanced (7 in controls, 4 mice in double treatment)? The number of mice should be the same or at least very close.

5-Figure 2E and 2F: are these the results of one experiment? Were the experiments repeated?

6-Figure 4A, caspase inhibitor ZVAD is used to assay apoptosis. The result is negative, but was the activity of ZVAD controlled to discard a technical issue?

7- Figure S1 panel E is supposed to be a measure of cell death in T- and B-cells. In healthy controls. Only one value per sample is presented. Are these the T cell values, the B cell values, or a mix of a mix of T and B cells? Why do they do not present both set of data?

Minor points:

8- Figure 1D and E: why was KG1 and not KG1a (as in Fig 1B) chosen for the retention experiment while the authors stated in the text that KG1a is the one that displays very potent drug efflux (this is the reason why it was used in Fig 1B).

9- Figure 2G needs careful editing, histograms are duplicated.

10- Edit 4.6 in Materials and Methods: a sentence finishes with the word “followed”. Either some words were deleted or the sentence needs editing/rewriting.

11- Conflicts of Interest: who is A.R.?

Author Response

Interesting study on the anti-leukemic effects of a combination therapy associating dendrogenin A (DDA) -a recently described anti-cancer agent- and an anthracycline (daunorobicin or idarubicin). The synergistic effects obtained using a novel agent with conventional drugs is a plus.  A similar manuscript describing the combination of DDA with cytarabine (ARA-C) has just been published in Cancers by the same authors. Although similar, this study brings enough novelty to be published independently. The molecular mechanisms described are of great interest in the field and beyond. There are however a few mandatory points that need to be addressed to justify the hypothesis and conclusions of the manuscript.

We thank the reviewer for his constructive and thoughtful comments concerning our manuscript. We have addressed all comments and concerns and hope to have improved our manuscript in a satisfactorily way.

Major points

1- Statistics were done using Mann-Whitney or t-test. This is Ok when comparing 2 sets of data, but many experiments include more than 2 sets of data. In those cases, these cannot be used. Use ANNOVA instead, otherwise, the multiple comparisons do not stand.

We have performed ANNOVA for multiple comparisons to avoid augmentation of alpha risk. Graphs have been modified accordingly.

2- Use classical standard deviation (SD) not SEM throughout the figures. Unlike SD, SEM is not a descriptive statistics (does not describe the variability of measures). SEM is only used here to minimize the error bar because SEM is smaller than SD).

 We have replaced SEM by classical standard deviation in all experiments, except experiment shown in panel 2F in order to keep it readable.

3- Figure 2E is a quantification of human MOLM14 cells in mice. The text and legend state that the blast cells were quantified in bone marrow and spleen. The figure should shows two quantification (bone marrow and spleen), why is there only one?

We are presenting the total tumor burden, hence the sum of blast cells in bone marrow and spleen, exactly as we did in our previously published papers (Ref: Saland, E.; Boutzen, H.; Castellano, R.; Pouyet, L.; Griessinger, E.; Larrue, C.; de Toni, F.; Scotland, S.; David, M.; Danet-Desnoyers, G.; et al. A robust and rapid xenograft model to assess efficacy of chemotherapeutic agents for human acute myeloid leukemia. Blood Cancer J 2015, 5, e297, doi:10.1038/bcj.2015.19. Farge, T.; Saland, E.; de Toni, F.; Aroua, N.; Hosseini, M.; Perry, R.; Bosc, C.; Sugita, M.; Stuani, L.; Fraisse, M.; et al. Chemotherapy-Resistant Human Acute Myeloid Leukemia Cells Are Not Enriched for Leukemic Stem Cells but Require Oxidative Metabolism. Cancer Discov 2017, 7, 716–735, doi:10.1158/2159-8290.CD-16-0441).

 We have now added the two separate quantifications that appear as supplementary data S1H and S1I.

4- Figure 2E: why are the groups imbalanced (7 in controls, 4 mice in double treatment)? The number of mice should be the same or at least very close.

Unfortunately, we realize that idarubicin was more toxic than we thought for NSG mice due to their defect of DNA strand break repair and one mice has to be sacrificed for ethical reason in the Idarubicin group and 3 in the combination therapy before the end of the experiment.

5-Figure 2E and 2F: are these the results of one experiment? Were the experiments repeated?

 The in vivo experiment 2F is the sum of two independent experiments. The CLDX experiment shown in panel 2E was only performed once, with a high number of mice. We believe this approach is sufficient to statistically prove that the normal environment of AML blasts (Bone Marrow or spleen) does not make them more resistant to the combination therapy.

6-Figure 4A, caspase inhibitor ZVAD is used to assay apoptosis. The result is negative, but was the activity of ZVAD controlled to discard a technical issue?

We have used z-VAD-FMK at the recommended concentration to inhibit PARP proteolysis and caspase 3 maturation. Moreover, these experiments were performed in similar conditions (timeframe and drug concentration) as in our recently published paper, where we found z-VAD-FMK to protect cells from cytarabine toxicity. Hence, we are convinced of the drug efficacy in these experimental conditions. Ref: Serhan, N.; Mouchel, P.-L.; Medina, P. de; Segala, G.; Mougel, A.; Saland, E.; Rives, A.; Lamaziere, A.; Despres, G.; Sarry, J.-E.; et al. Dendrogenin A synergizes with Cytarabine to Kill Acute Myeloid Leukemia Cells In Vitro and In Vivo. Cancers (Basel) 2020, 12, doi:10.3390/cancers12071725

7- Figure S1 panel E is supposed to be a measure of cell death in T- and B-cells. In healthy controls. Only one value per sample is presented. Are these the T cell values, the B cell values, or a mix of a mix of T and B cells? Why do they do not present both set of data?

These values represent a pool of T-cells (CD45+ CD3+) and B-cells (CD45+ CD19+). The goal was to evaluate the toxicity of the different drug regimens against different cell types present in normal blood,  especially myeloid and lymphoid cells. We tried to analyze B and T cells separately, but as B cells represented less than 10% of the sample, we felt it was informative to show results for the pooled populations.

Minor points:

8- Figure 1D and E: why was KG1 and not KG1a (as in Fig 1B) chosen for the retention experiment while the authors stated in the text that KG1a is the one that displays very potent drug efflux (this is the reason why it was used in Fig 1B).

KG1a is a subclone of KG1, but it’s very hard to transfect this cell line. We succeeded to transfect KG1, which express pgP at only slightly lower level than the sub-clone KG1a.

9- Figure 2G needs careful editing, histograms are duplicated.

Figure 2G has been edited.

10- Edit 4.6 in Materials and Methods: a sentence finishes with the word “followed”. Either some words were deleted or the sentence needs editing/rewriting.

Sentence has been edited.

11- Conflicts of Interest: who is A.R.?

It was a mistake, this has now been corrected (copy-paste typo from a previous version). However, we thank the reviewer for pointing this out , as we now acknowledge A.R. for providing us pure dendrogenin A.

Reviewer 2 Report

In this manuscript Mouchel et al. describe the anti-leukemic effect of Dendrogenin A. The manuscript is interesting and adds to the current limited knowledge on the effects of dendrogenin A in leukaemia.

I would advise the authors to explain in more detail the rationale of their experiments and add some references that could back up their choices in terms of biomarkers and assays.   The authors might want to provide further methodological details and molecular insights and resubmit the manuscript.

  • In Figure 1 the authors show the combination index of DDA + Daunorubicin. Can the author show the raw data and indicate which range of concentrations were tested? Can the author also give a brief description of the protocol in the methods?
  • In Figure 1 the authors demonstrate that the synergistic effect of DDA and Daunorubicin depends on the expression of DDA target LXR-B. Can the author confirm the efficiency of the knock down of LXR-B by qPCR or western blot? Or refer to a published paper where this was confirmed?
  • The experiments shown in Figure 1 and most of Figure 2 were performed by using trypan blue exclusion. In Figure 2, the same experiments, done this time on AML primary cells were instead performed by Annexin V staining. Can the authors explain why they used two different methodologies for cell lines and primary samples?
  • In Figure 3 the authors investigate the effect of DDA and anthracyclines on DNA damage. The fluorescence images are not visible. It is not possible to appreciate the 53BP1 foci. The authors might also want to explain why they preferred to use 53BP as DNA damage marker rather than yH2AX, an early marker of DNA damage and replicative stress.
  • In Figure 4 the author hypothesise that DDA in combination with Idarubicin induces authophagy. To prove their hypothesis the authors knock down VPS34. Can the authors explain their rationale and providence references for the role of VPS34 in autophagy?
  • In Figure 4C the authors indicate that the combination DDA+ idarubicin induces autophagy as shown by increase in levels of LC3-II. Can the author explain this marker and its role in autophagy? Can the authors add a reference? How do the authors explain that LC3-II increase mostly in OCI-AML3 but not in the other cell lines? Can multiple cell death mechanisms explain the anti-leukemic effect of their drug combination?

Author Response

In this manuscript Mouchel et al. describe the anti-leukemic effect of Dendrogenin A. The manuscript is interesting and adds to the current limited knowledge on the effects of dendrogenin A in leukaemia. I would advise the authors to explain in more detail the rationale of their experiments and add some  references that could back up their choices in terms of biomarkers and assays.   The authors might want to provide further methodological details and molecular insights and resubmit the manuscript.

We thank the reviewer for his comments concerning our manuscript. We have addressed all his concerns as outlined below.

In Figure 1 the authors show the combination index of DDA + Daunorubicin. Can the author show the raw data and indicate which range of concentrations were tested? Can the author also give a brief description of the protocol in the methods?

A brief description of the protocol indicating the tested range of drug concentrations has been added in the material and methods. A supplementary table has been made to show the combination index values we obtained.

In Figure 1 the authors demonstrate that the synergistic effect of DDA and Daunorubicin depends on the expression of DDA target LXR-B. Can the author confirm the efficiency of the knock down of LXR-B by qPCR or western blot? Or refer to a published paper where this was confirmed?

We have previously reported LXR-B knock-down efficiency by qRT-PCR and Western blot (Segala,G. et al, Nature Communications, 2017, supplementary figure 4I). We have controlled it again before performing these new experiments, and found comparable knock-down efficiency (Fig S2D). We have added this reference in the text and material and methods.  Ref: Segala, G.; David, M.; de Medina, P.; et al. Dendrogenin A drives LXR to trigger lethal autophagy in cancers. Nat. Commun. 2017, 8.

The experiments shown in Figure 1 and most of Figure 2 were performed by using trypan blue exclusion. In Figure 2, the same experiments, done this time on AML primary cells were instead performed by Annexin V staining. Can the authors explain why they used two different methodologies for cell lines and primary samples?

Annexin V staining was used as a secondary approach to control the results of trypan blue exclusion in cell lines. Due to the rarity of cells and the reasons outlined below, we performed only Annexin V staining in primary samples. Annexin V is one of the gold standard techniques to measure cell death. Moreover, flow cytometry was the only method allowing us to separate AML blasts from normal cells sometimes still present in blood of leukemic patients. We were thus able to evaluate the differences in toxicity against AML blasts (CD45+ CD33+ CD3- CD19-) or normal lymphoid cells (CD45+ CD33- CD3+ or CD19+). This allowed us to prove that the potentiation was selectively observed against AML blasts from primary samples (supplementary figure 1G).

In Figure 3 the authors investigate the effect of DDA and anthracyclines on DNA damage. The fluorescence images are not visible. It is not possible to appreciate the 53BP1 foci.

Figure has been resized to improve its quality. The foci, which are only present in the combination and in single Idarubicin treatment, are now more visible.

The authors might also want to explain why they preferred to use 53BP as DNA damage marker rather than yH2AX, an early marker of DNA damage and replicative stress.

γH2AX, the phosphorylated form of the histone variant H2AX, which is phosphorylated by the kinases ATR, ATM and DNA-PK in response to DSBs (double strand breaks), is not a direct marker of DSB and is known to spread not only around DSB but also single-strand breaks. This is the reason we preferred to quantify more precisely DSB by using a direct marker of DSB, the phosphorylated form of 53BP1 at Ser1778. Indeed, phosphorylation of this site has been shown to correlate with the amount of DSB. Ref : Lee JH, Cheong HM, Kang MY et al. Ser1778 of 53BP1 Plays a Role in DNA Double-strand Break Repairs. Korean J Physiol Pharmacol. 2009; 13(5):343±8.

In Figure 4 the author hypothesise that DDA in combination with Idarubicin induces authophagy. To prove their hypothesis the authors knock down VPS34. Can the authors explain their rationale and providence references for the role of VPS34 in autophagy?

VPS34 has been discovered in yeast and it forms at least two distinct macromolecular complexes, which separately regulate vacuolar protein sorting and macroautophagy. In mammalian, Vps34 controls autophagy via the production of PI(3)P.  Several papers have shown that autophagosome formation is completely blocked in the absence of Vps34 and the guidelines for the use and interpretation of assays for monitoring autophagy described VPS34 as an essential partner in the autophagy interactome. Knock-out or knock-down of VPS34 are very often used to study autophagy or its impact. Ref: Klionsky, D.J.; Abdalla, F.C.; Abeliovich, H.; Abraham, R.T.; Acevedo-Arozena, A.; Adeli, K.; Agholme, L.; Agnello, M.; Agostinis, P.; Aguirre-Ghiso, J.A.; et al. Guidelines for the use and interpretation of assays for monitoring autophagy. Autophagy 2012, 8, 445–544, doi:10.4161/auto.19496 ; We have already published that the knock down of VPS34 was able to block the autophagy induce by DDA. Ref: Segala, G.; David, M.; de Medina, P.; et al. Dendrogenin A drives LXR to trigger lethal autophagy in cancers. Nat. Commun. 2017, 8. Serhan, N.; Mouchel, P.-L.; Medina, P. de; Segala, G.; Mougel, A.; Saland, E.; Rives, A.; Lamaziere, A.; Despres, G.; Sarry, J.-E.; et al. Dendrogenin A synergizes with Cytarabine to Kill Acute Myeloid Leukemia Cells In Vitro and In Vivo. Cancers (Basel) 2020, 12, doi:10.3390/cancers12071725.

In Figure 4C the authors indicate that the combination DDA+ idarubicin induces autophagy as shown by increase in levels of LC3-II. Can the author explain this marker and its role in autophagy? Can the authors add a reference?

LC3 protein is a ubiquitin-like protein that can be conjugated to PE to form LC3-II.  LC3-II is the only protein marker that is reliably associated with completed autophagosomes. Guidelines for the use and interpretation of assays for monitoring autophagy (3rd edition) conclude that LC3 is the most widely monitored autophagy-related protein and is often an excellent marker for autophagic structures even if LC3 levels on their own do not address issues of autophagic flux.  For these reasons we decided for our paper to monitored LC3-II level. Ref: Klionsky, D.J.; Abdelmohsen, K.; Abe, A.; Abedin, M.J.; Abeliovich, H.; Acevedo Arozena, A.; Adachi, H.; Adams, C.M.; Adams, P.D.; Adeli, K.; et al. Guidelines for the use and interpretation of assays for monitoring autophagy (3rd edition). Autophagy 2016, 12, 1–222, doi:10.1080/15548627.2015.1100356.

This point and reference have been added to the manuscrit.

 How do the authors explain that LC3-II increase mostly in OCI-AML3 but not in the other cell lines?

There is an important variability in the basal level of autophagy flux. MOLM14 has a mutation in FLT3 that drives autophagy. For the three tested cell lines (MOLM-14,OCI-AML3 and KG1a), MOLM14 has the higher level of LC3-II, which explains the less important fold-increase in LC3-II level (fold change 2) compared to OCI-AML3 which has the lowest basal level of autophagy (fold change 14). Ref: Heydt, Q.; Larrue, C.; Saland, E.; Bertoli, S.; Sarry, J.-E.; Besson, A.; Manenti, S.; Joffre, C.; Mansat-De Mas, V. Ref: Oncogenic FLT3-ITD supports autophagy via ATF4 in acute myeloid leukemia. Oncogene 2018, 37, 787–797, doi:10.1038/onc.2017.376 ; Ref: Larrue, C.; Saland, E.; Boutzen, H.; Vergez, F.; David, M.; Joffre, C.; Hospital, M.-A.; Tamburini, J.; Delabesse, E.; Manenti, S.; et al. Proteasome inhibitors induce FLT3-ITD degradation through autophagy in AML cells. Blood 2016, 127, 882–892, doi:10.1182/blood-2015-05-646497.

Can multiple cell death mechanisms explain the anti-leukemic effect of their drug combination?

We have added the following text to answer this point: We previously showed that the combination of DDA, that induces cell death through lethal autophagy, with the antimetabolite ARAc, that induces mainly apoptosis, synergized in killing AML cells (Serhan et al, cancers 2020). Here, we show that the combination of DDA with a DNA damaging agent potentiates AML cell killing showing that multiple cell death mechanism improves the the yield of AML cells killing supporting the AL-lazikani's hypothesis (Al-Lazikani et al, Nat biotechnol, 2012). Since DDA has been proved to be efficient on various cell lines in vitro (de medina et al, J Med Chem, 2009; Segala et al, Nat commun, 2017; Bauriaud et al J steroid Biochem Mol Biol, 2019) and in vivo (de medina et al, Nat com, 2017; segala et al, Nat comm 2017; voisin et al, PNAS, 2017), this suggests that combination treatment of DDA with chemotherapeutic agents that kill cells through other mechanisms could be as well efficient and deserves further investigations.

Round 2

Reviewer 2 Report

The authors have addressed all my concerns. Please add all the relevant info and references to the manuscript.